# Natural Compounds of Marine Origin as Inducers of Immunogenic Cell Death (ICD): Potential Role for Cancer Interception and Therapy

**DOI:** 10.3390/cells10020231

**Published:** 2021-01-25

**Authors:** Clementina Sansone, Antonino Bruno, Concetta Piscitelli, Denisa Baci, Angelo Fontana, Christophe Brunet, Douglas M. Noonan, Adriana Albini

**Affiliations:** 1Stazione Zoologica Anton Dohrn, Istituto Nazionale di Biologia, Ecologia e Biotecnologie Marine, Villa Comunale, 80121 Napoli, Italy; clementina.sansone@szn.it (C.S.); concetta.piscitelli@szn.it (C.P.); christophe.brunet@szn.it (C.B.); 2Unit of Molecular Pathology, Biochemistry and Immunology, IRCCS MultiMedica, 20138 Milan, Italy; antonino.bruno@multimedica.it (A.B.); douglas.noonan@multimedica.it (D.M.N.); 3Department of Biotechnology and Life Sciences, University of Insubria, 211000 Varese, Italy; denisa.baci@gmail.com; 4Institute of Biomolecular Chemistry, National Research Council (CNR), 80078 Pozzuoli, Napoli, Italy; angelo.fontana@cnr.it; 5Department of Biology, University of Naples, 80126 Napoli, Italy; 6Laboratory of Vascular Biology and Angiogenesis, IRCCS MultiMedica, 20138 Milan, Italy

**Keywords:** immunogenic cell death, natural compounds, marine drugs, algae, prevention

## Abstract

Regulated cell death (RCD) has always been considered a tolerogenic event. Immunogenic cell death (ICD) occurs as a consequence of tumour cell death accompanied by the release of damage-associated molecular patterns (DAMPs), triggering an immune response. ICD plays a major role in stimulating the function of the immune system in cancer during chemotherapy and radiotherapy. ICD can therefore represent one of the routes to boost anticancer immune responses. According to the recommendations of the Nomenclature Committee on Cell Death (2018), apoptosis (type I cell death) and necrosis (type II cell death) represent are not the only types of RCD, which also includes necroptosis, pyroptosis, ferroptosis and others. Specific downstream signalling molecules and death-inducing stimuli can regulate distinct forms of ICD, which develop and promote the immune cell response. Dying cells deliver different potential immunogenic signals, such as DAMPs, which are able to stimulate the immune system. The acute exposure of DAMPs can prime antitumour immunity by inducing activation of antigen-presenting cells (APC), such as dendritic cells (DC), leading to the downstream response by cytotoxic T cells and natural killer cells (NK). As ICD represents an important target to direct and develop new pharmacological interventions, the identification of bioactive natural products, which are endowed with low side effects, higher tolerability and preferentially inducing immunogenic programmed cell death, represents a priority in biomedical research. The ability of ICD to drive the immune response depends on two major factors, neither of which is intrinsic to cell death: ‘Antigenicity and adjuvanticity’. Indeed, the use of natural ICD-triggering molecules, alone or in combination with different (immuno)therapies, can result in higher efficacy and tolerability. Here, we focused on natural (marine) compounds, particularly on marine microalgae derived molecules such as exopolysaccharides, sulphated polysaccharides, glycopeptides, glycolipids, phospholipids, that are endowed with ICD-inducing properties and sulfavants. Here, we discuss novel and repurposed small-molecule ICD triggers, as well as their ability to target important molecular pathways including the IL-6, TNF-α and interferons (IFNs), leading to immune stimulation, which could be used alone or in combinatorial immunotherapeutic strategies in cancer prevention and therapies.

## 1. Introduction

Cancer incidence and mortality are expected to increase with aging population, underpinning the need for prevention and treatments. Conventional cancer therapies aim to kill and permanently eliminate tumour cells from the organism; however, they can induce severe side effects on healthy cells and multiple organ dysfunction, often with chronic consequences [1,2,3]. Off-target toxicities are exerted on the cardiovascular system, neural cells, liver, kidney, bone marrow and other organs [1,2,3]. One of the most common side effects of chemotherapy are the nonspecific antiproliferative activities of the anticancer drugs on leukocytes and lymphocytes, which can induce a suppression of the immune system with a consequent higher susceptibility to infections [3].

Cancer is characterized by uncontrolled cell proliferation and/or evasion of cell death [4]. Dysregulation of the regulated cell death program (RCD) represents one of the strategies adopted by neoplastic cells to strengthen and enhance their growth [5,6]. Programmed cell death (PCD) represents a noninflammatory cellular process, allowing us to eliminate aged or damaged cells, maintaining tissue homeostasis [7].

Homeostatic and pathogen-driven (non-self-cell death and antigenic) are two forms of cell death [8], and the ‘self’ vs. ‘non-self’ recognition model represents a key factor in cancer diseases. In the past, cell death has been classified as apoptosis (nonimmunogenic and physiologically regulated) and in necrosis (pathological, uncontrolled and immunogenic) [9]. Recent evidence has demonstrated that this dichotomy is no longer accepted [10], and that other diverse types of nonapoptotic PCD exist, including necroptosis, pyroptosis, ferroptosis, entotic cell death, entotic cell death, parthanatos, lysosome-dependent cell death, autophagy-dependent cell death, alkaliptosis and oxeiptosis [11,12,13,14]. Methuosis is one of the most recent updates within the family of nonapoptotic cell death phenotypes, which is characterized by the cytoplasmic accumulation of single-membrane vacuoles in cells and by the loss of the attachment from the neighbouring cells [15,16]. Defence from pathogens and homeostasis regulation often induce cell death through the activation of a genetically encoded molecular machinery, which can trigger an antigen-specific immune response. From this observation, a new conceptualization of cancer cell death includes immunogenic pathways, involving antigenicity (the ability to induce an immune cell response) for the engagement of antigen-specific immune responses.

Besides antigenicity, another crucial factor stimulating an effective immune response is the adjuvanticity (the ability of potentiating the immune cell response), which can be conferred by pathogens- and/or danger-associated molecular patterns (PAMPs and DAMPs, respectively). Molecules are released by dying cells, including DAMPs and PAMPs stimulating and the recruitment and maturation of antigen-presenting cells (APCs) [6,17]. This ‘alarmin’ or adjuvanticity state is sensed by pattern recognition receptors (PRRs), present in innate immune cells such as monocytes, macrophages and dendritic cells (DCs), thus promoting their activation and maturation and to stimulate the adaptive arm of the immune system [18]. ‘The Nomenclature Committee on Cell Death’ [5] has recently defined immunogenic cell death (ICD) as: ‘A form of regulated cell death, that is sufficient to activate an adaptive immune response in immunocompetent syngeneic hosts’, which properly reflects the two major components of ICD as a process, that is, the cellular component and the host component. Importantly, the latter does not refer to potential defects of the host that prevent the initiation of adaptive immunity (e.g., HLA mismatch, systemic immunodeficiency), but to features intrinsic to dying cells that render them immunogenic only in specific hosts. ICD is therefore defined as when death is able to promote an immune response through the production and/or release of immunomodulatory molecules [19] or antigenicity, as well as the stimulation of several immune system-induced pathways [5,20,21,22] or adjuvanticity. The two events, antigenicity and adjuvanticity, are not necessarily occurring together.

Dying cancer cells are able to stimulate different immune cells by inflammatory, chemical mediators, and DAMPs, which thus represent the connection between cell death induction and immune response [14], or they can deliver tolerogenic signals that suppress the immune response [23]. Many of the ICD pathways have a key role in the induction of antitumour immunity [24], becoming of interest for biomedical research [25]. Cancer therapies should preferably target the ICD response, as it exerts a cytotoxic effect on the neoplastic tissues and enhances the immune system for a broad antitumour immunity.

The biological activity, structural diversity and variety of mechanisms of action of natural products [26] are unrivalled with designed synthetic libraries or chemical scaffolds [27,28]. In addition, discovery related to ICD might take advantage of the knowledge originating from traditional medicine, with the support of modern ethno-pharmacology. Currently, some cancer therapies are known to be able to activate the release of DAMPs, thus promoting the immune system response against cancer development [5,29,30]. Amongst these molecules, most are of synthetic origin and fewer are natural, i.e., extracted or reproduced from living organisms [31,32]. It is therefore of great interest to look for new compounds/new organisms (species, families) able to enhance and/or diversify the immune function.

The discovery of the immune checkpoint molecules, such as cytotoxic T lymphocyte-associated protein 4 (CTLA-4), programmed cell death 1 (PD-1) receptor and its ligand PD-L1, has led to the rapid development of therapeutic approaches aimed at restoring and re-educating the altered/aberrant host immune cell response, by stimulating immune cells of the host [33,34]. The use of immune checkpoint inhibitors, mostly antibodies which block the ligand–receptor interactions, induces reactivation of key immune cell functions and has been demonstrated to have great clinical benefits in several tumours [33,34]. Recent preclinical and clinical data suggest that the localization, quality and quantity of lymphoid and myeloid cells within the tumor microenvironment play a major role in shaping the response to an immune checkpoint blockade. Despite the strong clinical success of cancer immunotherapy with checkpoint inhibitors, most patients still do not experience a durable response (13) and many do not respond at all, and there might be hyperprogressors with these drugs [35].

The marine environment offers a high variety of systems (environmental and biological) bringing a high level of biodiversity (mostly unexploited or unknown), and a boundless chemodiversity. Thus far, less than twenty drugs, based on compounds of marine origin (and macro-organisms derived), are approved for clinical purposes, mainly synthetic derivatives of the original natural molecules. Among them, six are used in cancer therapy: Cytarabine (Ara-C), eribulin mesylate (E7389), trabectedin (ET-743), brentuximab vedotin (SGN-35), polatuzamab vedotin and aplidin [36,37]. In addition, many other compounds are currently under clinical investigations, between the Phase II and Phase III of clinical trials, with promising anticancer activities [38].

Marine micro-organisms are now receiving particular attention in this context, also because they are more easily exploitable and expandable than marine macro-organisms [39,40]. Marine microalgae could be considered one of the richest sources of known and unknown bioactive compounds on our globe, that can be developed as chemopreventive and anticancer agents or functional foods [41]. Marine microalgae have already shown antitumour/anti-proliferative properties [42], demonstrating applied research advantages [43]. Some of them are able to induce activation of death signalling pathways in vitro on diverse human cancer cell lines [44], probably exerting immuno-modulatory activities, proposing them as a potential candidate as ICD inducer drugs for anti-cancer therapies [45,46].

Here, we review and discuss the current knowledge of the ICD process in cancer, reporting the diversity and effects of the ICD-inducing molecules used so far, with a special emphasis on natural chemical compounds. We focus on microalgal compounds and on studies displaying their impact on the immune system, as well as their ability to target important molecular pathways including the IL-6, TNF-α and interferons (IFNs) leading to immune stimulation. We aim to stress the potential of microalgae derivatives as ICD-inducers and/or stimulators the immunomodulatory pathway, to foster research on microalgae for immune-related cancer therapies.

## 2. Immunogenic Cell Death and Cancer

The mammalian immune system is crucial for the defence and elimination of infectious pathogens and damaged cells through specific recognition mechanisms and signalling systems with factors secreted from the cells in the extracellular matrix [47].

During tumour propagation, there is an increase in metabolic demands leading to metabolic, genetic, hypoxic and/or mechanical stress, which can trigger cell death. Three major different profiles of cancer cell death have been recently described: Tolerogenic cell death (TCD), inflammatory cell death and ICD [48]. Cancer cells can activate the TCD, a homeostatic or physiological cell death that causes an increased production of anti-inflammatory factors, resulting in the suppression of anticancer immunity. Simultaneous tolerogenic and immunogenic signals can cause the activation of biological processes, such as inflammatory cell death [49]. The balance between the inflammatory and tolerogenic immune response has to be constantly regulated, and its imbalance can lead to carcinogenesis through different mechanisms, including DNA damage caused by inflammation-associated reactive oxygen and/or nitrogen species and resistance to cell death through activation of survival factors, such as NF-κB [50], which is a major regulator of inflammation.

There are two principal ways for cancers to escape the immune response: A lack of immunogenicity, due to cancer cells’ low rates of antigenicity and/or the suppression of antitumour host defence. ICD plays a crucial role in the induction of antitumour immunity, by associating programmed cell death (PCD) pathways, death factors, immune signals, DAMPs and the immune response (Figure 1). The complexity is even higher as ICD develops though different cell death programs (Figure 1).

The key mechanism triggering antitumour immune responses, especially during ICD, involves the recruitment and activation of dendritic cells (DC). The induction of an adaptive immune response begins when DCs first engulf fractions of cancer cells or interact with cancer cells promoting ICD. This process leads to the formation of tumour-associated antigens (TAAs) that will determine DC maturation and differentiation with secretion of pro-inflammatory cytokines, such as IL-1β, IL-12 and TNF-α. Following migration, through lymphatic vessels to lymph nodes, DCs deliver antigens to naïve T lymphocytes that instruct differentiation in CD8^+^ cytotoxic and CD4^+^ helper T cells [51]. Activated Th cells proliferate and secrete additional cytokines, producing a cascade effect on B cells, which are licensed to produce specific antibodies aimed to eliminate pathogens or damaged cells. Together with shaping T cell responses, DCs release cytokines, such as IL-12, IL-18, IL-15 and IFNγ, that promote NK cell activation and their cytolytic activity. Fragment (Fc) region interaction with CD16 on NK cells can stimulate the release of antibody-dependent cellular cytotoxicity (ADCC) [52]. Moreover, DCs contribute to the stimulation of memory B lymphocytes, which produce specific antibodies in response to a second exposure to the same antigens [53,54,55].

Evidence suggests that agents inducing ICD could potentially use the dying cancer cells as ‘vaccines’ to reactivate immune surveillance, through the maturation of DCs and activation of CTLs [56], as well as enhancing the cytotoxic activity of NK cells, generating a more efficient antitumour response, prognosis and improved patient survival. Cancer cell death induced by some chemotherapies are known to cause the extracellular release of DAMPs-based signals, such as calreticulin (CRT), adenosine triphosphate (ATP), high-mobility group box 1 protein (HMGB1) and heat shock protein 70 (HSP70) [49,57], which can favour inflammatory processes, including type I IFNs, and the activation of innate immune cells (DCs, macrophages, neutrophils and NK cells) and the downstream recruitment of adaptive immune cells (e.g., T and B cells) [49,57]. In addition, the efficacy of anticancer therapies might also focus on the achievement of a balance between ICD and TCD, to specifically enhance the immune system response against neoplastic cells and harmful signals contributing to the onset of tumours.

## 3. Anticancer Drugs and Terrestrial Natural Compounds Inducing ICD

Several anticancer drugs, such as cyclophosphamide, oxaliplatin, 5-fluorouracil and mitoxantrone, induce ICD through different molecular mechanisms. Cyclophosphamide has been found to elicit an antitumour response by directly favouring the expansion of NK and T cells in transplantable murine glioma [58] and lymphoma [59]. Studies on mouse models show that cyclophosphamide also promotes an antitumour immune response, inducing the delocalization of Gram-positive bacteria of the intestinal microbiota to secondary lymphoid organs, via gap junctions, in the intestinal epithelium, stimulating the production of Th17 cells and the secretion of IL-17 and IFNγ [60,61]. Oxaliplatin stimulates ICD by inducing the ER stress-dependent exposure of calreticulin (CARL), as shown in a murine colorectal carcinoma [62]. CARL, a biochemical signal of apoptosis, instructs DCs to engulf the apoptotic cells [6]. Oxaliplatin can also promote the activation of cytotoxic T lymphocytes in both transgenic and transplantable murine models of prostate cancers [63]. 5-Fluorouracil increases the frequency of tumour-infiltrating cytotoxic T lymphocytes in colorectal cancers, activating HMGB1 (High-Mobility Group Box 1) and ATP secretion [64,65]. HMGB1 is released by dying cells into the extracellular environment, during the late stages of PCD, positively regulates autophagy and it binds to different receptors on DCs, thus promoting antigen presentation [6,66]. In addition, the release of ATP is a key event of ICD, by recruiting DCs and promoting the activation of the NLRP3 inflammasome, via P2X7 receptors and the proteolytic production of IL-1β [67]. Moreover, ATP can be converted into ADP, which binds distinct receptors inducing immunosuppression [68]. The mitoxanthrone can induce exposure of CARL in human colorectal cancer cells [69,70] and the autophagic process via the release of ATP and HMGB1 from necrotic cells in pancreatic and breast cancer cells [71].

One of the major challenges of cancer therapy research is to find novel natural compounds able to elicit antitumour effects through ICD induction [24]. This challenging aim is even more important, considering that ICD not only inhibits primary tumours, but dramatically suppresses distant metastatic tumours [72], thus exerting an abscopal effect (*‘ab scopus’*, away from the target), useful in cancer therapy. Daunorubicin and doxorubicin (Table 1) belong to the anthracycline glycoside family, which was among the first chemical families to be documented as anticancer drugs with ICD properties. These compounds were characterised for the first time from *Streptomyces peucetius*, a microorganism belonging to phylum of Actinobacteria [73]. Daunorubicin and doxorubicin are the active principles of two antibiotics interacting with polydeoxyribonucleotides and DNAs, by the intercalation between the base pairs of native DNA, causing DNA damage and single-strand breaks. Doxorubicin induces apoptotic cell death in a caspase-dependent manner in many cancer cell types, such as colorectal carcinoma and melanoma murine cells (IC_50_: 25 µM for CT26 cells, 30 µM for Pro-B cells and 2.5 µM for B16-F10 cells) with the consequent activation of immune response mediated by DCs and CD8^+^ T-cells [74]. Caspase inhibition by Z-VAD-fmk (inhibitor of the catalytic site of caspase proteases, 100 µM) drastically reduced the immunogenicity of dying tumour cells treated with doxorubicin in several rodent models of neoplasia, with a specific effect on DC maturation and DC-mediated recognition and phagocytosis [74].

A similar immunogenic effect was also reported on other cell lines treated with anthracyclines, such as acute myeloid leukaemia (AML) [75] and neuro-2a neuroblastoma cells [86]. Treatment of AML with daunorubicin (2 and 4 µM) induced CARL exposure at the cell surface and HSP70/HSP90 release (2 and 4 ng mL^−1^ after 18 h, respectively). CD8α^+^ T-cells co-cultured with doxorubicin-treated neuroblastoma cells became responsive to anti-CD3/CD28 antibody stimulation, with a consequent increased proliferation rate and augmented IFNγ release.

Polyphenols, one of the most abundant secondary metabolite families, are natural compounds with antiproliferative effects against many cancer cell types in vitro, and have also been proposed as ICD inducer molecules [77]. As an example, a gallotannin-rich fraction obtained from *Caesalpinia spinosa* displayed an antiproliferative effect on melanoma cells, with an IC_50_ of 63.5 μg mL^−1^ on B16-F10 and of 70.1 μg mL^−1^ on A375 cells [76]. The anticancer effect of gallotannin is promoted through apoptosis markers of caspases 3 and 9, mobilization of cytochrome C and externalization of annexin V. In addition, a gallotannin-rich fraction from *Caesalpinia spinosa* induces the expression of ICD markers (ATP and HMGB1) and activation of the autophagic process [76]. This study reported that a gallotannin-rich fraction (101.6 µg mL^−1^ for 48 h) induced 70% of cell death in lysate of melanoma cells (B16-F10), being highly immunogenic as mice vaccinated with cell lysate were able to drastically reduce the tumour volume of injected B16-F10 cells [76].

Linalool and p-coumaric acid are other interesting molecules, providing an efficient strategy to contrast tumour expansion. These compounds were able to inhibit the growth of many human cancer cells, such as A549 adenocarcinoma cells, T-47D breast cancer cells, SW620 colon adenocarcinoma cells and Hep G2 liver cancer cells, in a dose-dependent way [77]. In addition, these two polyphenols induced the secretion of pro-inflammatory cytokines in lymphocytes, such as IFN-γ, IL-13, IL-2, IL-21, IL-21R, IL-4, IL-6sR and TNF-α [77]. Shikonin, another example of a phenolic compound isolated from the Chinese herbal medicine *Lithospermum erythrorhizon* [78], is an efficient adjuvant molecule for the activation of ICD in cancer cells. Melanoma murine cancer cells (B16-F10) treated with shikonin activated both apoptotic death mechanisms, receptor- and mitochondria-mediated pathways, through caspase 8, Bax, cytochrome c and caspase 9 [78]. In addition, the cell lysate enhanced the maturation of DCs and acted as differentiation stimuli for Th1 and Th17 cells [78,79], which enhances the presentations of the ICD antigens.

Capsaicin (8-methyl-N-vanillyl-6-nonenamide), an alkaloid extracted from plants of the genus *Capsicum* that gives the spicy flavours to hot peppers, has repellent activities against mammals and fungi [87]. Capsaicin is able to selectively induce cell death in many types of human cancer cells, such as breast cancer cells MDA-MB-231 and MCF-7, urothelial bladder cancer 5637 cells, BxPC-3 and AsPC-1 pancreatic cancer cells, SNU-1 and TMC-1 gastric cancer cells, and SW480 and HCT-116 colorectal cancer cells, without lowering viability in normal cells [80]. Capsaicin activates both vanilloid receptor 1 (VR1)-dependent and -independent pathways, promoting cell death through Reactive Oxygen Species (ROS) generation and endoplasmic reticulum stress [82]. Capsaicin-mediated cell death supports the expression of hallmarks of ICD. Capsaicin induces apoptotic cell death in PEL cells (primary effusion lymphoma), favouring translocation of CARL and HSP90 to the cell surface [88].

CARL, HSP70 and HSP90 and ATP release were exposed at the cell surface during the early apoptotic stage also of human bladder cancer cells (SD48 and T24) treated with 50–250 µM of capsaicin, ICD markers [81].

Digoxin and digitoxin are saponins (glycosylated sterols) isolated from the plants *Digitalis lanata* and *Digitalis purpurea*, respectively, ouabain and lanatoside C (extracted from *Acokanthera schimperi* or *Strophanthus gratus*), and lanatoside C (*Digitalis lanata*). All these molecules are strong ICD inducers, as shown on a large panel of human cancer cells, such as U2OS osteosarcoma cells, HeLa cervical adenocarcinoma cells, HCT 116 colon adenocarcinoma cells, A549 nonsmall cell lung carcinoma cells, LNCap prostate carcinoma, Cal27 oral squamous carcinoma, HepG2 hepatocellular carcinoma cells and MDA-MB 231 breast adenocarcinoma cells [83,84,85]. The overall survival of carcinoma patients receiving digoxin was superior as compared with that of control patients, and the subgroup analyses revealed that digoxin enhanced the overall survival of breast, head and neck, hepatocellular and colorectal carcinoma patients [84,85].

## 4. Marine Microalgal Compounds Inducing ICD in Cancer Cells

Microalgae can be a relevant source of different classes of compounds with potential interest for human health. Several applications as pharmaceuticals, nutraceuticals and food supplements products are possible. There is much evidence regarding the antiproliferative and anticancer activities of microalgal-derived compounds and immunomodulatory activity; however, there is still little data on ICD induction. The plethora of the in vitro studies performed on the human cancer cell lines often highlighted that microalgal extracts, fractions and compounds activate specific cell death signalling pathways (Table 2). 

Recently, it was demonstrated that microalgal species such as *Spirulina* sp. when provided as a food supplement are endowed with immunomodulating activity (Table 2) [101]. Sulfavants are a family of synthetic sulfoglycolipids mimicking natural α-sulphoquinovosides of the diatom *Thalassiosira weissflodgi* [27,102,103]. The family prototype Sulfavant A is a potent activator of DC maturation at micromolar concentration [102]. Unlike any other available molecule, this sulfolipid does not induce production of pro-inflammatory cytokines but triggers the upregulation of MHC II and co-stimulatory molecules, including CD83, CD86 and CD54, which are necessary for the differentiation of naïve T cells. Sulfavant A is currently under preclinical tests as a vaccine adjuvant and its efficacy has already been proven in a murine model of a melanoma vaccine [102]. Although the product does not show any cytotoxic activity, mice treated with the vaccine containing Sulfavant A did not show progression of the tumour for more than 10 days when B16F10 melanoma cells were injected subcutaneously to the animals. The effect is potentially exerted through the immunomodulation together with vaccination, and it is suggested to derive from the enhancement of the cytotoxic response of the immune system against the tumour cells.

A glycopeptide isolated from the marine dinoflagellate *Alexandrium minutum* (Table 2) is able to activate a mitophagic cell death in the cancer cell line without affecting normal cell line viability [45]. This form of microautophagy causes the lysosomal secretion of ATP, which recruits myeloid cells [104] and promotes an immunogenic sequential cell death cascade.

Astaxanthin from the microalga *Haematococcus pluvialis* potentially mediates the suppression of VEGF in vascular endothelial cells (Table 2) and stimulates the release of IFN-γ and IL-12, mediating the cytokines storm in the case of inflammation [105]. IFN-γ is one of the key factors for DCs maturation and macrophage activation, thus favouring cell immunity response inhibiting angiogenesis and metastasis formation. Further investigations are required to clarify the specific pathway induced by astaxanthin, already in use for neurodegenerative diseases prevention.

The polyunsaturated aldehydes (PUAs) isolated from three diatoms, *Thalassiosira rotula*, *Skeletonema costatum* and *Pseudonitzschia delicatissima* (Table 2), are able to induce specific PCD, extrinsic apoptosis and necroptosis in lung and colon adenocarcinoma cell lines [44,91]. Necroptosis can be initiated by immune ligands, such as Fas, TNF superfamily receptors and CD40, which activate the receptor-interacting protein kinase 3 (RIPK3) [106,107,108]. In turn, RIPK3 (and MLKL) causes the release of ATP and HMGB1, which are known as ICD inducers [109].

Polysaccharides and sulphated polysaccharides from marine origin activate NF-ĸB, and produce TNF-α, IL-6, INF-γ and TLR-4 expression in monocytes cell lines in vitro [110]. A broad range of species (*Alexandrium tamarense, Chaetoceros calcitrans, Skeletonema costatum, Dunaliella salina, Euglena gracilis, Skeletonema marinoi, Thalassiosira weissflogii, Spirulina maxima, Tribonema sp and Skeletonema dohrni*) belonging to different algal classes (Table 2) are able to induce IL-6 release in PBMC [111], an important step in the immune system activation. IL-6 released by DCs, together with other cytokines like IL-1β, IL-12 or TNF, shapes the natural killer cell (NK) and T cell responses [93], an immunomodulatory role. This feature is even more important considering that cancer diseases tend to develop mechanisms of immunosuppression to evade anti-cancer immune responses, as for example, preventing cytotoxic T lymphocytes (CTLs) or NK cells from reaching and killing tumour cells or by polarizing immune cells to acquire pro-angiogenic activities [112,113,114,115,116].

Similar activity is exerted by the exopolysaccharide, p-KG03, isolated from *Gyrodinium impudicum*, and by sulphated polysaccharides, from *Tribonema* sp. (Table 2). In particular, the compound p-KG03 is able to induce IFN-γ and IL-2 release, targeting preferentially NK cells with the specific activation against host neoplastic cells [117]. The sulphated polysaccharides from *Tribonema* sp. activate the macrophages through release of cytokines (IL-6, IL-12), which are the main path for ICD activation [110].

The compound coibamide from *Leptolyngbya* induces autophagy in glioblastoma cells through a ULK-phosphorylation (Table 2). Autophagic cell death is responsible for the DAMPs release, with a consequential activation of both inflammatory response and immune system cells for the elimination of cell debris [118].

The clinical trials of the marine compounds that have cancer activity (treatment and/or chemoprevention) are in Table 3.

The gut microbiome has a significant influence on the local and systemic immune system [119,120,121]. Damping and enhancing the immune systems for the composition of the gut microbiome is another possible effect of marine derivatives. The gut microbiome can influence the outcome of immune checkpoint blockade therapy in preclinical mouse models and humans, and it can strongly influence the outcome of cancer patients receiving checkpoint blockade therapy [122,123,124,125]. Two microalgal compounds are able to induce microbiota modulation isolated from *Spirulina maxima* and *Schizochytrium* sp. (Table 2). It was recently demonstrated that the modified pectin from *Spirulina maxima* modulates the gut microbiota in mice, inducing mucin, IFN-α and IL-6 release, key factors for ICD activation during an inflammatory cascade [96]. Similarly, the polyunsaturated fatty acids (PUFAs) from *Schizochytrium* modulate lymphocyte function by targeting plasma membrane molecular organization such as physical properties (compressibility, phospholipid flip-flop, acyl chain packing, elasticity) and chemical properties (lipid domain formation) [126]. The polyunsaturated fatty acids from *Schizochytrium* induce microbiota stimulation and mucin release [97]. In recent studies, it was demonstrated how PUFAs disrupt membrane domain organization through changes in lipid rafts activating specific signalling networks through the microbiota stimulation and cytokines release [127], and anti-inflammatory activities [128].

## 5. Conclusions

Besides synthetic anti-cancer chemotherapeutics-targeted anticancer agents and cyclin kinase inhibitors, there are also natural products that are able to induce ICD. Natural compounds from plant origin acting as ICD inducers on tumour cells could represent a new frontier in cancer interception and therapy. Among them, molecules of marine origin represent very attractive new compounds for ICD. In this scenario, microalgae are particularly promising, as confirmed by recent pre-clinical studies. Here, we reviewed ICD, some terrestrial plants-derived ICD inducers, and presented microalgae as future promising agents to generate novel moieties, endowed with ICD and/or immunomodulating activities. The aim is to pave the way for further research on this topic, to develop preclinical and clinical trials for candidate microalgae compounds as ICD inducers for cancer (co)treatments, cancer prevention, interception and therapy.

## Figures and Tables

**Figure 1 cells-10-00231-f001:**
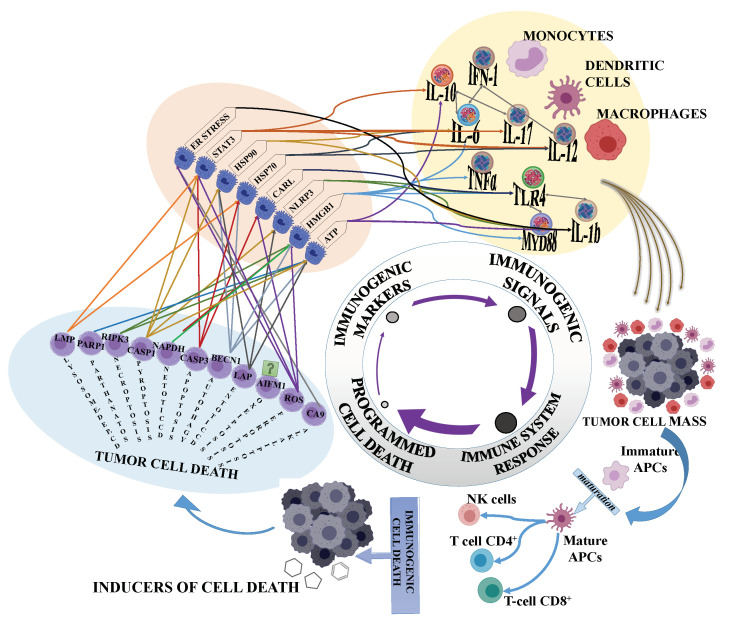
From programmed cell death to immunogenic stimulation. Specific cell death pathways instruct the production of immunogenic markers that in turn stimulate innate immune cells (macrophages, dendritic cells and monocytes) to enhance immune anticancer response, through priming and differentiation of adaptive immune cells (CD4^+^, CD8^+^ and B cells), as well as natural killer cells (NK) cells. The combination and cooperation of cell death and immunogenic stimuli could contribute to increase the therapeutic success to eliminate tumour cell mass.

**Table 1 cells-10-00231-t001:** Natural compounds and relative immunogenic cell death (ICD)-related molecular mechanisms. Selected natural compounds and related ICD activities are shown, clustered for tumour cell type target and ICD properties. (sp. is species)

Source	Compound or Fraction	Tumour Cell Type	ICD Pathway	Refs
*Streptomyces peucetius* (bacteria)	Doxorubicin	CT26 colon carcinoma cell line Pro-B melanoma murine cell line B16-F10 melanoma cell line	-DC maturation -Involvement of CD8+ T-cells	[74]
*Streptomyces peucetius* (bacteria)	Daunorubicin	AML acute myeloid leukemia cells	-CARL exposure -HSP70/HSP90 release -IFNγ release	[75]
*Caesalpinia spinose* (higher plant)	Gallotannin-rich Fraction	B16-F10 melanoma cell line A-375 melanoma cell line	-Activation of caspases 3 and 9 -ATP and HMGB1	[76]
*Plantago sp.* (higher plant)	Linalool and p-coumaric	A549 lung carcinoma cell line T-47D breast cancer cell line SW620 colon adenocarcinoma cell line Hep G2 liver cancer c cell line	-Release of pro-inflammatory cytokines: IFN-γ, IL-13, IL-2, IL-21, IL-21R, IL-4, IL-6sR and TNF-α	[77]
*Lithospermum erythrorhizon* (higher plant)	Shikonin	B16-F10 melanoma cell line	-DC maturation -Differentiation stimuli for Th1 and Th17 cells	[78,79]
*Capsicum sp.* (higher plant)	Capsaicin	MDA-MB-231 breast cancer cell line MCF-7 breast cancer cell line 5637 urothelial bladder cancer cell line T24 bladder cancer cell line SD48 bladder cancer cell line BxPC-3 pancreatic cancer cell line AsPC-1 pancreatic cancer cell line SNU-1 gastric cancer cell line TMC-1 gastric cancer cell line SW480 colorectal cell line cancer cell line HCT-116 colorectal cancer cell line Primary effusion lymphoma cells	-ROS generation -Endoplasmic reticulum stress -CARL exposure -HSP70/HSP90 release -HMGB1 release -ATP	[80,81,82]
*Digitalis sp.* (higher plant)	Digoxin Digitoxin Lanatoside C Ouabain	U-2 OS osteosarcoma cell line and other tumour cells lines	-CARL exposure -ATP and HMGB1 release	[83,84,85]

**Table 2 cells-10-00231-t002:** Marine organisms and relative immunomodulating molecular mechanisms. Selected marine organisms and related immunomodulating activities are shown, clustered for tumour cell type target and immunomodulating properties.

Source	Compound or Fraction	Cell Type Target	ICD and Immune Activation	Refs
***Alexandrium minutum*** (dinophyceae)	Glycopeptide	A549 Lung adenocarcinoma cell line	Mitophagy	[45]
***Hematococcus pluvialis*** (green alga)	Astaxanthin	Primary lymphocytes	IFN-γ and IL-2 release	[89,90]
***Thalassiosira rotula*****, *Skeletonema costatum*** **and *Pseudonitzschia delicatissima*** (diatoms)	Polyunsaturated aldehydes (PUAs)	Caco-2Colon adenocarcinoma cell line A569 Lung adenocarcinoma cell line COLO 205 Colon adenocarcinoma cell line	Extrinsic apoptosis Necroptosis	[44,91]
***Alexandrium tamarense*** (dinophyceae)	Acetonitrile/aqueous fraction (Glycolipids/phospholipids)	Human Peripheral Blood Mononuclear Cell (PBMC)	IL-6 release	[92,93]
***Chaetoceros calcitrans*** (diatoms)	Aqueous fraction (amino acids/saccharides)	Human Peripheral Mononuclear Blood Cell (PBMC)	IL-6 release	[92,93]
***Skeletonema costatum*** (diatoms)	Methanolic extract (Apolar compounds) Dichloromethane/ethanol fraction (Triglycerides)	Human Peripheral Mononuclear Blood Cell (PBMC)	IL-6 release	[92,93]
***Dunaliella salina*** (green alga)	Methanol/aqueous fraction (nucleosides)	Human Peripheral Mononuclear Blood Cell (PBMC)	IL-6 release	[92,93]
***Euglena gracilis*** (euglenophyceae)	Paramylon	Human Peripheral Blood Mononuclear Cell (PBMC)	IL-6 and TNF-α release	[94]
***Gyrodinium impudicum*** (dinophyceae)	Exopolysaccharide, p-KG03	Lymphocytes Natural killer (NK)	IFN-γ and IL-2 release	[95]
***Skeletonema marinoi*** (diatoms)	Methanolic extract (Apolar compounds)	Human Peripheral Blood Mononuclear Cell (PBMC)	IL-6 release	[5]
***Thalassiosira weissflogii*** (diatoms)	Glycolipids and Phospholipids	Human Peripheral Blood Mononuclear Cell (PBMC)	IL-6 release upregulation of MHC II, CD83, CD86, CD54	[92,93]
***Spirulina maxima*** (cyanophyceae)	Modified pectin (SmP)	Modulation of gut microbiota	Mucin, IFN-α, IL-6 release	[96]
***Schizochytrium sp.*** (Labyrinthulea)	Polyunsaturated fatty acids	Modulation of gut microbiota	Lymphocytes target	[97]
***Thraustochytriidae sp***. (Labyrinthulea)	Exopolysaccharides	Antibodies production stimulation	B-cell proliferation	[98]
***Tribonema sp.*** (Xanthophyceae)	Sulphated polysaccharides	Macrophages	Cytokines release (IL-6, IL-12)	[99]
***Skeletonema dohrni*** (diatoms)	Methanol/aqueous fraction (nucleosides)	Human Peripheral Blood Mononuclear Cell (PBMC)	IL-6 release	[92,93]
**Leptolyngbya** **(cyanophyceae)**	Coibamide	U87-MG Human glioblastoma cells cell line	ULK phosphorylation, autophagy activation	[100]

**Table 3 cells-10-00231-t003:** Summary of relevant clinical trials employing the marine compounds endowed with anti-cancer activity (treatment and/or chemoprevention).

	Status Clinical Trials. Gov Identifier:	Study Title	Conditions	Interventions	Locations
**Peptidoglycan**	Recruiting NCT04183478	The Efficacy and Safety of K-001 in the Treatment of Advanced Pancreatic Cancer	Pancreatic Cancer	Drug: K-001 (K001 is peptidoglycan, prepared from the fermentation of the marine microorganism Spirulina) Other: Placebo	RenJiH Shanghai, Shanghai, China
**Fucoidan**	Recruiting NCT04066660	Study of Oligo-Fucoidan in Advanced Hepatocellular Carcinoma (HCC)	Advanced Hepatocellular Carcinoma	Dietary Supplement: Oligo Fucoidan Dietary Supplement: Placebo	Fudan University Zhongshan Hospital Shanghai, China
	Recruiting NCT04597476	A Randomized, Double-blind Study to Evaluate the Clinical Effect and Safety of Fucoidan in Patients with Squamous Cell Carcinomas of the Head and Neck	Squamous Cell Carcinomas of the Head and Neck	Dietary Supplement: Fucoidan Other: Placebo (Potato starch)	National Taiwan University Hospital Taipei county, Taiwan
	Recruiting NCT04342949	The Auxiliary Effects of Fucoidan for Locally Advanced Rectal Cancer Patients	To Observe Whether the Fucoidan Can Improve the Quality of Life of Such Patients Receiving the Neoadjuvant CCRT	Behavioural: Quality of life	Chung-Ho Memorial Hospital, Kaohsiung Medical University: Kaohsiung, Taiwan
**Product from Red Marine** **Algae**	Recruiting NCT03869905	Aquamin^®^ as an Adjuvant Intervention for Ulcerative Colitis	Ulcerative Colitis	Drug: Aquamin^®^ Drug: Placebo first then Aquamin^®^ Aquamin^®^, a Multi-mineral Natural Product from Red Marine Algae, as an Adjuvant Intervention for Mild Ulcerative Colitis and Ulcerative Colitis in Remission	The University of Michigan Ann Arbor, Michigan, United States
**AMR101 Marine oil**	Active, not recruiting NCT04216251	PRevention Using EPA Against coloREctal Cancer	Colorectal Adenoma Colorectal Cancer	Drug: AMR101 (VASCEPA, icosapent ethyl)	Massachusetts General Hospital Boston, Massachusetts, United States
	Recruiting NCT03661047	OMega-3 Fatty Acid for the Immune Modulation of Colorectal Cancer	Colon Cancer	Drug: AMR101 (VASCEPA, icosapent ethyl)	Massachusetts General Hospital Boston, Massachusetts, United States
**Marine oil**	Active, not recruiting NCT04269876	A Study to Evaluate the Effects of a Marine Lipid Oil Concentrate Formulation on Inflammation	Inflammation Inflammatory Response	Dietary Supplement: Marine Lipid Oil Concentrate Dietary Supplement: Dietary Supplement Dietary Supplement: Placebo	Lfie Extension Clinical Research, Inc. Fort Lauderdale, Florida, United States
	Recruiting NCT04209244	Effect of Fish Oil on Hyperlipidemia and Toxicities in Children and Young Adults with Acute Lymphoblastic Leukemia	Leukaemia, Acute Lymphoblastic	Dietary Supplement: Eskimo-3 Pure Fish Oil Dietary Supplement: Rapeseed Oil	Aalborg University Hospital Aarhus University Hospital Aarhus, Rigshospitalet Copenhagen, Odense University Hospital Denmark
	Completed Has Results NCT01661764	Fish Oil Supplementation, Nutrigenomics and Colorectal Cancer Prevention	Colorectal Adenomatous Polyps	Drug: Eicosapentanoic acid and docosahexanoic acid Drug: Oleic Acid	Vanderbilt University Medical Center Nashville, Tennessee, United States
	Completed Has Results NCT01813110	Effects of a Prescription Omega-3 Fatty Acid Concentrate on Induced Inflammation	Inflammatory Responses	Drug: 4 g prescription omega-3 concentrate Drug: Placebo	Penn State University Pennsylvania United States
**VITAL: Marine oil and Vitamin D**	Unknown † NCT02239874 † Study has passed its completion date and status has not been verified in more than two years	VITamin D and OmegA-3 TriaL: Effects on Mammographic Density and Breast Tissue	Benign Breast Disease	Dietary Supplement: Vitamin D and fish oil placebo Dietary Supplement: Fish oil and vitamin D placebo Dietary Supplement: Vitamin D placebo and fish oil placebo Dietary Supplement: Vitamin D and fish oil	Brigham and Women’s Hospital Boston, Massachusetts, United States
	Active, not recruiting Has Results NCT01169259	Vitamin D and Omega-3 Trial (VITAL)	Cancer Cardiovascular Disease	Dietary Supplement: Vitamin D3 Drug: Omega-3 fatty acids (fish oil) Dietary Supplement: Vitamin D3 placebo Dietary Supplement: Fish oil placebo	Brigham and Women’s Hospital Boston, Massachusetts, United States
	Active, not recruiting NCT04386577	Effects of Vitamin D and Omega-3 Supplementation on Telomeres in VITAL	Aging	Dietary Supplement: Vitamin D3 (cholecalciferol) Drug: Fish oil	Georgia Prevention Institute Augusta, Georgia, United States
**Trabectedin**	Recruiting NCT03886311	Talimogene Laherparepvec, Nivolumab and Trabectedin for Sarcoma	Sarcoma	Drug: Talimogene Laherparepvec 100000000 PFU/1 ML Injection Suspension [IMLYGIC] Drug: Nivolumab IV Soln 100 MG/10 ML Drug: Trabectedin 0.25 MG/1 VIAL Intravenous Powder for Solution	Sarcoma Oncology Center Santa Monica, California, United States
	Completed NCT02249702	Activity of Trabectedin or Gemcitabine + Docetaxel in Uterine Leiomyosarcoma	Leiomyo sarcoma	Drug: Gemcitabine + docetaxel Drug: Trabectedin	Centro di Riferimento Oncologico Aviano, Pordenone, Italy 33 institutes
**Eribulin** **mesylate**	Terminated NCT01534455	Efficacy and Tolerability of Eribulin Plus Lapatinib in Patients with Metastatic Breast Cancer (E-VITA)	Metastatic Breast Cancer	Drug: Lapatinib + 1.23 mg Eribulin Drug: Lapatinib + 1.76 mg Eribulin	Klinikum der Otto-v.-Guericke-Universität Frauenklinik Magdeburg, Germany

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
