# Peer review of "Natural Compounds of Marine Origin as Inducers of Immunogenic Cell Death (ICD): Potential Role for Cancer Interception and Therapy"

_cells, 2021, doi:10.3390/cells10020231_

Round 1

Reviewer 1 Report

The manuscript from Sansone et al. aims at giving an overview of immunogenic cell death(ICD)-inducing compounds, with a special focus on microalgae-derived compounds. Marine compounds are a relevant source of anticancer drugs, and there is a growing interest in their function as immune-modulators. Thus, an updated survey of the microalgae derived-compounds able to induce immune-modulation or trigger ICD is of potential interest. However, the manuscript in its present form does not fulfill its aims and needs substantial editing and reorganization. In general, there is a lack of precision and clearness in the terminology used throughout the text. Moreover, the focus should be revised based on the reported evidences. Please consider the following:

  • In the abstract at line 22, the authors state that “some stress-induced RCD can drive an inflammatory or immune response, leading to immunogenic cell death (ICD)”. This statement is somehow confusing. ICD happens when there is cancer cell death, accompanied by release of DAMPS, triggering an immune response (PMID:30955159), whereas the verb "leading" in the reported sentence argues the opposite. Please rephrase.
  •  
  • Abstract, line 24, the authors state: “Necrosis and Apoptosis are only part of the RCD…”. The correct term would be “types”, instead of “part”. Moreover the term “necrosis, without further indication, historically refers to an accidental form of cell death and not a regulated one. Even if RCD can present with morphological necrotic features, the sentence as it is is  misleading. The authors should look in detail and refer to the indications of the Nomenclature Committee for Cell Death, throughout the paper. (PMID: 29362479)
  • Asbtract, line 29: “Chronic Acute exposure of DAMPS”. Is it chronic or acute?
  •  
  • Confusion exists throughout the paper about ability to induce ICD and the ability to activate immune cells. The two events are not necessarily occurring together: cytokine release from immune cells or other markers of immune activation can be stimulated upon treatment with several compounds without the concomitant induction of ICD. See also wrong statements at lines 341-342, 354-355. Several marine compounds have been reported to exert immune modulating effects (mainly in in vitro assays) without evidences of ICD being reported alongside (PMID: 32070764).

  • The introduction exposes in a sometimes not effective manner some basic concepts. Lines 71-92 are particularly hazy: Line 80: open inverted commas are not closed afterwards. Line 73 and 74: please explain the terms “antigenicity” and “adjuvanticity” and their relevance within the focus of the review. Lines 124-125: correct “marine microorganisms”. Lines 129-132: This sentence describes the ICD-inducing activity of microalgae-derived compounds as “probable” and “potential”. This is in contrast with the description present in the manuscript title, in the abstract, and in the title of paragraph 4. These all should be changed to better adapt the aims of the manuscript to the evidences reported in paragraph 4.

  • Paragraph 2, lines 141-163: the authors describe again types of cell death providing further nomenclature: please put together with the first description and homogenize.

  • Line 184. Please correct “our refs” and put correct references.

  • Line 188-190. The concept of DAMP release is here repeated. If one main interest is ICD induction, this concept should be made clear earlier, with description of specific DAMPs.

  • Paragraph 4 is reporting evidences on microalgae-derived compounds, which is the particular focus of the review. The two parts of the manuscript (paragraphs 1-3 and paragraph 4) are not well-balanced in term of space.

  • Line 229: please delete “SPOSTARE TABELLA”.

  • Line 218-220 somehow too strong, please contextualize. Please define abscopal effect induction.

  • Paragraph 4: no clear direct evidence of induction of DAMPS/markers of ICD by the listed compound in a given model is reported, thus the paragraph is not wellmatching with the previous sections. The description mostly reports on immune-modulating activities of the listed compounds. The authors should separate the two aspects, and extend the description to other marine compounds with immune-modulating activities, to give an updated overview of the state of the art in this field and provide new insights respect to existing review work (PMID: 31861368).
  •  
  • Table content should be adapted based on the previous observations.

  • Line 301, reference format is wrong

  • Line 328-330, please use italics wherever appropriate

  • Line 349 please define ICB

Author Response

The point-by-point responses to reviewer’s comments.

REVIEWER1

Open Review

English language and style

( ) Extensive editing of English language and style required
(x) Moderate English changes required
( ) English language and style are fine/minor spell check required
( ) I don't feel qualified to judge about the English language and style

Is the work a significant contribution to the field?        **

Is the work well organized and comprehensively described? **

Is the work scientifically sound and not misleading? **

Are there appropriate and adequate references to related and previous work? **

Is the English used correct and readable?   ***

Comments and Suggestions for Authors

  • The manuscript from Sansone et al. aims at giving an overview of immunogenic cell death(ICD)-inducing compounds, with a special focus on microalgae-derived compounds. Marine compounds are a relevant source of anticancer drugs, and there is a growing interest in their function as immune-modulators. Thus, an updated survey of the microalgae derived-compounds able to induce immune-modulation or trigger ICD is of potential interest. However, the manuscript in its present form does not fulfill its aims and needs substantial editing and reorganization. In general, there is a lack of precision and clearness in the terminology used throughout the text. Moreover, the focus should be revised based on the reported evidences. Please consider the following:

We thank the reviewer for these observation and comments. In our revised version, precision and clearness in the terminology is addressed throughout the text. Moreover, the focus is revised based on reported evidence.

  • In the abstract at line 22, the authors state that “some stress-induced RCD can drive an inflammatory or immune response, leading to immunogenic cell death (ICD)”. This statement is somehow confusing. ICD happens when there is cancer cell death, accompanied by release of DAMPS, triggering an immune response (PMID:30955159), whereas the verb "leading" in the reported sentence argues the opposite. Please rephrase. 

We rephrased the sentence according to the reviewer’s comment and literature. Reference “Arch Pharm Res. 2019 Jul;42(7):629-645. doi: 10.1007/s12272-019-01150-z. Epub 2019 Apr 6. Natural compound inducers of immunogenic cell death by Marc Diederich, has been added.

We clarified when cancer cell death, accompanied by release of DAMPS, triggers an immune response, we corrected the word “leading”.

  • Abstract, line 24, the authors state: “Necrosis and Apoptosis are only part of the RCD…”. The correct term would be “types”, instead of “part”.

We rephrased the sentence according to the reviewer’s comment and literature.

  • Moreover, the term “necrosis, without further indication, historically refers to an accidental form of cell death and not a regulated one. Even if RCD can present with morphological necrotic features, the sentence as it is misleading. The authors should look in detail and refer to the indications of the Nomenclature Committee for Cell Death, throughout the paper. (PMID: 29362479)

We revise the sentence according to the reviewer’s comment Nomenclature has been used according to “Review Cell Death Differ. 2018 Mar;25(3):486-541. doi: 10.1038/s41418-017-0012-4. 2018, Molecular mechanisms of cell death: recommendations of the Nomenclature Committee on Cell Death 2018 suggested reference.

  • Abstract, line 29: “Chronic Acute exposure of DAMPS”. Is it chronic or acute? 

We apologize for the lack of clarity. Acute is the correct term.

  • Confusion exists throughout the paper about ability to induce ICD and the ability to activate immune cells. The two events are not necessarily occurring together: cytokine release from immune cells or other markers of immune activation can be stimulated upon treatment with several compounds without the concomitant induction of ICD. See also wrong statements at lines 341-342, 354-355. Several marine compounds have been reported to exert immune modulating effects (mainly in in vitro assays) without evidence of ICD being reported alongside (PMID: 32070764).

We thank the reviewer for these observation and comments. In our revised version, for the molecule discussed, we differentiate between their ability to induce ICD and the ability to activate immune cells. We stressed when these two events occur together.

We cited the “Review Semin Cancer Biol. 2020 Feb 15;S1044-579X(20)30041-9. doi: 10.1016/j.semcancer.2020.02.008. Immune-modulating and anti-inflammatory marine compounds against cancer by Cristina Florean, Mario Dicato, Marc Diederich.

  • The introduction exposes in a sometimes not effective manner some basic concepts. Lines 71-92 are particularly hazy: Line 80: open inverted commas are not closed afterwards.

We exposed basic concepts as a general overview, to make our review readable and accessible to non-immunologist; to respond to the reviewer we made it more effective. Line 80 has been fixed

  • Line 73 and 74: please explain the terms “antigenicity” and “adjuvanticity” and their relevance within the focus of the review.

The two terms have been explained: Antigenicity refers to the ability to induce an immune cell response through the engagement of antigen-specific immune responses. Adjuvanticity refers to the ability of potentiating the immune cell response, which can be conferred by pathogens- and/or danger- associated molecular patterns, by the use of “adjuvant-like” molecules.

  • Lines 124-125: correct “marine microorganisms”. Lines 129-132: This sentence describes the ICD-inducing activity of microalgae-derived compounds as “probable” and “potential”. This is in contrast with the description present in the manuscript title, in the abstract, and in the title of paragraph 4. These all should be changed to better adapt the aims of the manuscript to the evidences reported in paragraph 4.

The sentence describing the ICD-inducing activity of microalgae-derived compounds has been fixed. “Probable” and “potential” has been removed. We have made the suggested changes to better adapt the aims of the manuscript to the evidence reported in paragraph 4.

  • Paragraph 2, lines 141-163: the authors describe again types of cell death providing further nomenclature: please put together with the first description and homogenize.

We thank the reviewer for the comment. We deleted redundant sections. The description of cell death has been put together in the dedicated section.

  • Line 184. Please correct “our refs” and put correct references.

The corrected reference has been added.

  • Line 188-190. The concept of DAMP release is here repeated. If one main interest is ICD induction, this concept should be made clear earlier, with description of specific DAMPs.

We described the concept of DAMP release in a single session, not to repeat it, the DAMPs involved have been detailed with related references.

  • Paragraph 4 is reporting evidence on microalgae-derived compounds, which is the particular focus of the review. The two parts of the manuscript (paragraphs 1-3 and paragraph 4) are not well-balanced in term of space.

We reorganized the paragraph content and length, to make them more balanced. Some redundant sections, that made the manuscript uneven, have been removed and repetition avoided.

  • Line 229: please delete “SPOSTARE TABELLA”.

The sentence has been deleted.

  • Line 218-220 somehow too strong, please contextualize. Please define abscopal effect induction.

We softened the concept in line 218-220. The term “abscopal” referred as 'ab scopus', away from the target was explained in the main text.

  • Paragraph 4: no clear direct evidence of induction of DAMPS/markers of ICD by the listed compound in a given model is reported, thus the paragraph is not wellmatching with the previous sections. The description mostly reports on immune-modulating activities of the listed compounds. The authors should separate the two aspects, and extend the description to other marine compounds with immune-modulating activities, to give an updated overview of the state of the art in this field and provide new insights respect to existing review work (PMID: 31861368). 

Paragraph 4 has been extensively revised to separate the two aspects. This made the section more consistent with its content. Other marine compounds with immune-modulating activities have been listed. Reference suggested by the reviewer, Mar Drugs. 2019 Dec 18;18(1):2. doi: 10.3390/md18010002. Microalgae with Immunomodulatory Activities by Gennaro Riccio, Chiara Lauritano has been integrated and discussed in paragraph 4.

  • Table content should be adapted based on the previous observations.

The tables are now consistent.

  • Line 301, reference format is wrong

We run back the EndNote library. The reference is now in the correct format.

  • Line 328-330, please use italics wherever appropriate

Corrected

  • Line 349 please define ICB

The acronym has been deleted (ICB), we used immune checkpoint blockade three times.

Reviewer 2 Report

The authors bring a very sound review article covering natural compounds originating from marine organisms, which are able to induce immunogenic cell death and they discuss their role in cancer treatment. I am convinced, that nowadays, this topic is a very hot one and will attract a large number of readers and the article reads very nicely. Nevertheless, some issues should be improved before the publication of this article.

Behind the words “natural products” in line 32 add the following reference:
J Med Chem 2020 Mar 12;63(5):1937-1963. doi: 10.1021/acs.jmedchem.9b01509. Epub 2020 Feb 17.
Sarco/Endoplasmic Reticulum Calcium ATPase Inhibitors: Beyond Anticancer Perspective

Line 122 and 123 – it would be very beneficial for the reader if a table listing the compounds in the clinical trials + indication + state of the clinical trial would be added

Line 124: “Marine microorganisms are now receiving a particular attention in this context, also because more easily exploitable and expandable than marine microorganisms”

I do not understand the sentence, it is a cycle – marine micoroog. are more exploitable than marine microorg.? I do not understand

Latin words must be in Latin (e.g. in vitro, but also others)

Line 133 – unify the sentence tenses

Figure 1: the stuff in the image (both cartoons and the text) is totally unreadable, even when magnified to a large extent, you must definitely redo this and also provide an image of much better resolution, otherwise, it is meaningless, and there is no added value of the image

Table 1 – the structures of the compounds must be added to the table, otherwise it is quite meaningless

Also, in the text, you should definitely discuss the structure-activity relationship, otherwise, it is only a list of activities…

“Tumor cell type” in table 1 – I recommend you to check the proper name of the cell lines according to the ATCC database, very often, it is imprecise and not properly written

Regarding the last row in the table, there are definitely many more cardiac glycosides inducing ICD, you should either list all or write cardiac glycosides in general, I would prefer listing all of them, otherwise, it is just random selection, if so, you must justify why, why these two (in clinics, they are not doing very well – narrow biological window, etc. and even for drug repositioning, novel CG candidates are being looked for, etc.)

Line 234 – what do you mean by ICD-relied

Line 235 – a typo in the word compound

Line 235 – “are showed” – wrong English

Table 2 – why the table does not follow the same format in all columns as table 1, first compounds, second source, etc., also cell abbreviation first, then cell line explanation, why it is written differently? I would recommend to write only cell line abbreviations and explain them either in the table caption/legend or in a footnote, it will be much more clear and better arranged

In general, it would be better to focus only on pure compounds (plus showing the structure and discussing MOA and SAR), since in extracts it can be the synergic effect of more compounds and after identification, usually, none of them alone exerts the desired effect

The compound and extract selection for this review seems quite random

Moreover, sentences including phrases as: “activates many factors” and similar are empty and it seems like the authors were lazy to list the involved factors in total

Author Response

REVIEWER2

Open Review

English language and style

( ) Extensive editing of English language and style required
(x) Moderate English changes required
( ) English language and style are fine/minor spell check required
( ) I don't feel qualified to judge about the English language and style

Is the work a significant contribution to the field?

Is the work well organized and comprehensively described? ****

Is the work scientifically sound and not misleading? ***

Are there appropriate and adequate references to related and previous work? ****

Is the English used correct and readable? ****

Comments and Suggestions for Authors

  • The authors bring a very sound review article covering natural compounds originating from marine organisms, which are able to induce immunogenic cell death and they discuss their role in cancer treatment. I am convinced, that nowadays, this topic is a very hot one and will attract a large number of readers and the article reads very nicely. Nevertheless, some issues should be improved before the publication of this article.

We thank the reviewer for this very positive comment, about bringing a very sound review article covering natural compounds originating from marine organisms able to induce ICD.

  • Behind the words “natural products” in line 32 add the following reference:
    J Med Chem 2020 Mar 12;63(5):1937-1963. doi: 10.1021/acs.jmedchem.9b01509. Epub 2020 Feb 17. Sarco/Endoplasmic Reticulum Calcium ATPase Inhibitors: Beyond Anticancer Perspective.

Reference suggested has been added.

  • Line 122 and 123 – it would be very beneficial for the reader if a table listing the compounds in the clinical trials + indication + state of the clinical trial would be added

A table listing clinical trials (clinicaltrials.gov) with marine compounds that have cancer as a target has been assembled (Table 3).

  • Line 124: “Marine microorganisms are now receiving a particular attention in this context, also because more easily exploitable and expandable than marine microorganisms”

Sentence has been corrected.

  • I do not understand the sentence, it is a cycle – marine micoroog. Are more exploitable than marine microorg.? I do not understand

It was a typo; the second word is macroorganisms. The phrase is now: “marine microorganisms are more exploitable than marine macroorganisms”, we amended that.

  • Latin words must be in Latin (e.g. in vitro, but also others)

We checked all the text for consistency.

  • Line 133 – unify the sentence tenses

Tenses have been unified.

  • Figure 1: the stuff in the image (both cartoons and the text) is totally unreadable, even when magnified to a large extent, you must definitely redo this and also provide an image of much better resolution, otherwise, it is meaningless, and there is no added value of the image

In figure 1, we have enlarged the fonts and made higher resolution, they are readable now, the cartoons have been expanded as well.

  • Table 1 – the structures of the compounds must be added to the table, otherwise it is quite meaningless Also, in the text, you should definitely discuss the structure-activity relationship, otherwise, it is only a list of activities…

Table 1 include molecules whose structures are well known or molecules mixed extracts; therefore, adding structure to the table would be uneven; we prefered to point out to the cellular targets and activities. We agree that to discuss structure-activity relationship of the molecules selected would be relevant. We feel that this will deserve a more “chemical paper” paper, for a structural chemistry/biophysical audience.

  • “Tumor cell type” in table 1 – I recommend you check the proper name of the cell lines according to the ATCC database, very often, it is imprecise and not properly written

We checked on ATCC for cell line names. We also make Tab 1 and Tab 2 more consistent, in the way they are presented.

  • Regarding the last row in the table, there are definitely many more cardiac glycosides inducing ICD, you should either list all or write cardiac glycosides in general, I would prefer listing all of them, otherwise, it is just random selection, if so, you must justify why, why these two (in clinics, they are not doing very well – narrow biological window, etc. and even for drug repositioning, novel CG candidates are being looked for, etc.)

We thank the reviewer for this suggestion. We listed the cardiac glycosides inducing ICD and provided the related references.

  • Line 234 – what do you mean by ICD-relied

Corrected with ICD-related

  • Line 235 – a typo in the word compound

Corrected

  • Line 235 – “are showed” – wrong English

Corrected

  • Table 2 – why the table does not follow the same format in all columns as table 1, first compounds, second source, etc., also cell abbreviation first, then cell line explanation, why it is written differently? I would recommend writing only cell line abbreviations and explain them either in the table caption/legend or in a footnote, it will be much more clear and better arranged

We make Table 1 and Table 2 consistent in the way they are presented.

  • In general, it would be better to focus only on pure compounds (plus showing the structure and discussing MOA and SAR), since in extracts it can be the synergic effect of more compounds and after identification, usually, none of them alone exerts the desired effect

Extracts are generally more active than single compounds. Exactly because they can be synergic therefore, we presented both pure compounds and extracts, which are more suitable for nutraceutical applications.

  • The compound and extract selection for this review seems quite random

We apologized if this is the impression. We tried to discuss our selection more clearly. We selected the compounds discussed as the most relevant example to their described activities. The choice of the compounds of marine origin was driven by the aim of stressing the relevance of the marine source (still strongly under-estimated) to obtain molecules/extracts, endowed with beneficial effects on human health.

  • Moreover, sentences including phrases as: “activates many factors” and similar are empty and it seems like the authors were lazy to list the involved factors in total.

We thank the reviewer for this observation. We abolished empty phases and we detail sections that might have been too broad in their statements.

Round 2

Reviewer 1 Report

Please substitute "ICD" with "immunomodulating" at the following lines:

296, 297, 388.

Author Response

Please substitute "ICD" with "immunomodulating" at the following lines:

296, 297, 388.

We made the changes.

Reviewer 2 Report

The authors have reflected on all suggestions and comments of the reviewer. The article can be accepted for publication.

Author Response

(The authors gave the same response as above.)
